# Mechanistic insights into tropical circulation and hydroclimate responses to future forest cover change

Nora L. S. Fahrenbach<sup>1</sup>, Robert C. J. Wills<sup>1</sup>, and Steven J. De Hertog<sup>2</sup>

<sup>1</sup>Institute for Atmospheric and Climate Science, ETH Zurich, Zurich, Switzerland

**Correspondence:** Nora L. S. Fahrenbach (nora.fahrenbach@env.ethz.ch)

Abstract. Afforestation and the prevention of deforestation are important climate mitigation strategies, alongside reductions in greenhouse gas emissions. However, the biogeophysical effects of potential future forest cover change on the atmospheric circulation and tropical hydroclimate remain uncertain. We address this research gap using future scenario simulations from seven multi-ensemble models participating in the Land Use Model Intercomparison Project (LUMIP). The largest scenario differences in afforestation and avoided deforestation are located in the tropics, leading to robust increases in local evapotranspiration and precipitation, but widespread decreases in net precipitation (precipitation minus evapotranspiration), especially over Africa. Our results suggest that two competing mechanisms shape the tropospheric circulation and net precipitation response over Africa: Not only do forests increase evaporation, but they also increase surface momentum fluxes, thereby slowing near-surface winds and reducing orographic net precipitation. Opposing this surface drag effect is an energetic effect due to increased net energy input to the atmosphere, which strengthens convection and increases net precipitation. While the surface drag effect dominates and leads to a net precipitation decrease over western and southeastern Africa, the energetic effect dominates and leads to a net precipitation increase over central Africa. This tropical hydroclimate response to the forest cover change is largely independent of the background climate under low- to medium-warming scenarios. Our findings contribute to an improved understanding of the mechanisms of forest cover impact on future hydroclimate changes in the tropics and highlight the importance of considering hydroclimatic feedbacks in the context of future afforestation strategies.

#### **Short summary**

Afforestation is a key strategy for climate change mitigation, yet the impacts on tropical hydroclimate remain uncertain. We find that potential future afforestation would increase evaporation and precipitation in the tropics, especially over Africa. However, it would reduce net precipitation (precipitation minus evaporation), which determines water availability. This happens because trees slow near-surface winds, while their influence on the energy budget would otherwise strengthen convection.

<sup>&</sup>lt;sup>2</sup>Q-ForestLab, Department of Environment, Ghent University, Belgium

#### 1 Introduction

30

55

Afforestation and the prevention of deforestation are widely recognized as key strategies for climate change mitigation by enhancing carbon sequestration, alongside efforts to reduce greenhouse gas emissions (Roe et al., 2021; Girardin et al., 2021; Seddon et al., 2021). All scenarios in the IPCC AR6 report which limit global warming to 2°C or less by 2100 rely on carbon dioxide removal (CDR) in addition to emissions reductions (IPCC, 2022). Nature-based solutions such as afforestation are increasingly valued for their cost-effectiveness compared to technological CDR methods, like direct air carbon capture (DACCS) (Griscom et al., 2017). Moreover, forest expansion and conservation provide additional co-benefits, including biodiversity preservation, improved water quality, and enhanced societal well-being (Chausson et al., 2020; Seddon, 2022).

The cooling due to forests' influence on the carbon and other biogeochemical cycles can be either enhanced or partially to fully offset by biogeophysical effects—climate responses to changes in surface albedo, evaporative efficiency, and surface roughness—depending on the location of forestation (Pongratz et al., 2021; Bonan, 2008; Claussen et al., 2001; Windisch et al., 2022). Increasing tree cover generally reduces surface albedo compared to grass- or cropland, leading to greater absorption of shortwave solar radiation by the surface (Bonan, 2002). Afforestation in boreal regions thus leads to regional biogeophysical warming due to the large albedo reduction, particularly during winter and spring when trees mask the snow-covered ground (Bonan et al., 1992; Betts, 2000). However, trees also enhance evapotranspiration through their larger leaf area and deeper root systems (Bonan, 2008), physiological control of transpiration through canopy conductance, as well as through the enhancement of turbulent fluxes by their influence on surface roughness. This is the dominant mechanism by which afforestation in tropical regions induces regional biogeophysical cooling (Claussen et al., 2001; Davin and De Noblet-Ducoudré, 2010). In addition to contributing to enhanced evaporation, the influence of trees on surface roughness increases turbulent fluxes in general, including momentum fluxes, which can weaken wind speeds (Burakowski et al., 2018; Sud et al., 1988). These biogeophysical effects interact with atmospheric processes and in turn impact regional to global energy and moisture fluxes, leading to impacts on cloud formation, precipitation, and surface runoff (Swann et al., 2012; Liang et al., 2022; Devaraju et al., 2015).

Previous studies have examined how forest cover changes influence local precipitation patterns, with findings that vary based on forest patch size, local climate, or specific atmospheric mechanisms (e.g., Boysen et al., 2020; Luo et al., 2022; Khanna et al., 2017; Smith et al., 2023; Qin et al., 2025). Large-scale deforestation has been consistently linked to reduced local precipitation in modelling studies due to reduced local moisture recycling, and vice versa for afforestation (Boysen et al., 2020; Luo et al., 2022; Spracklen et al., 2018; te Wierik et al., 2021). However, uncertainties in precipitation responses remain large, primarily due to model differences in moisture advection and evapotranspiration responses, alongside internal climate variability (Boysen et al., 2020; Luo et al., 2022). Observational evidence from satellite and reanalysis data corroborates these modeling results, demonstrating that historical pan-tropical deforestation (particularly at scales exceeding 50 km) has led to significant precipitation declines (Smith et al., 2023), and that the recent increase in global vegetation cover has induced precipitation increases, especially in humid regions (Zeng et al., 2018). Unlike the precipitation decline observed with large-scale deforestation, smaller deforestation patches can lead to an increase in local precipitation, for instance, due to enhanced mesoscale convection following reduced surface roughness (Baidya Roy and Avissar, 2002; Khanna and Medvigy, 2014). These findings

could imply that precipitation increases in response to deforestation at small scales may transition into precipitation decreases at larger scales (Khanna et al., 2017).

Beyond local impacts, afforestation can induce significant remote precipitation changes by modifying atmospheric circulation and moisture transport (Swann et al., 2012; Portmann et al., 2022; Laguë et al., 2021). In particular, extratropical afforestation in the Northern Hemisphere has been shown to induce latitudinal shifts in tropical precipitation through its impact on the energy budget, leading to a northward displacement of the Intertropical Convergence Zone (ITCZ), and vice versa for deforestation (Swann et al., 2012; Portmann et al., 2022; Liang et al., 2022; Devaraju et al., 2015). These ITCZ shifts, in turn, influence precipitation patterns across the global monsoon regions (Liang et al., 2022; Devaraju et al., 2015). However, changes in forest cover within the tropics generally exert stronger local than remote effects (Devaraju et al., 2015), similar to other forcings like aerosols (Allen et al., 2015).

Only a few studies have focused on how afforestation can alter the balance between precipitation (P) and evapotranspiration (E), and consequently affect net precipitation (P-E), runoff, and water availability (Cui et al., 2022; Zan et al., 2024; Lejeune et al., 2015). Net precipitation plays an important role in land-atmosphere interactions by regulating the moisture exchange between the surface and the atmosphere. Critically, if evapotranspiration were to increase without any accompanying changes in atmospheric moisture transport, precipitation would increase by the same magnitude through conservation of moisture, resulting in net-zero changes in P-E. Therefore, to explain any non-zero changes in P-E, there must be changes in atmospheric moisture transport. Thus, changes in P-E are closely linked to atmospheric circulation changes, which drive large-scale redistribution of moisture and energy (e.g., Seager et al., 2010; Ma and Xie, 2013; Chadwick et al., 2013; Wills et al., 2016). Most previous studies about P-E responses to forest cover changes have not accounted for these large-scale circulation adjustments since they relied on atmospheric moisture tracking based on the climatological circulation (Cui et al., 2022; Zan et al., 2024). As a result, their conclusions regarding afforestation-induced moisture fluxes are likely incomplete since the leading-order influence of circulation on P-E is missing. The key knowledge gap, therefore, is understanding how atmospheric circulation responds to future forest cover changes and how this, in turn, shapes regional and global hydroclimate. Addressing this gap is essential given the importance of forestation as a future climate mitigation strategy.

To address this knowledge gap, we analyse hydroclimate changes in the tropics in future afforestation simulations from the Land Use Model Intercomparison Project (LUMIP; Lawrence et al., 2016). We particularly focus on local hydroclimate changes over Africa, which is projected to experience the most significant forest cover and climate changes in the LUMIP simulations but remains understudied compared to the Amazon rainforest. We aim to address the following fundamental research questions: (1) What are the biogeophysical impacts of future afforestation and avoided deforestation on the atmospheric circulation and tropical hydroclimate (P, E, P - E)? (2) What are the underlying physical mechanisms driving these local hydrological responses? Can these effects be attributed to changes in surface drag, albedo, and evaporation? Answering these questions will advance our mechanistic understanding of land use and land cover change (LULCC) impacts on the hydrological cycle and will improve the understanding of the potential consequences of afforestation scenarios that help reach net-zero emissions.

#### 2 Data and methods

#### 90 2.1 LUMIP simulations

We analyse changes in tropical hydroclimate to future LULCC, particularly forest cover changes, using simulations from the Land Use Model Intercomparison Project (LUMIP; Lawrence et al., 2016):

- SSP126-SSP370Lu (hereafter S1L3): This concentration-driven experiment was run with all forcings identical to the Shared Socioeconomic Pathway SSP1-2.6 (global radiative forcing of 2.6 Wm<sup>-2</sup> by 2100; hereafter S1) except that the LULCC input is taken from SSP3-7.0.
- SSP370-SSP126Lu (hereafter S3L1): This concentration-driven experiment was run with all forcings identical to SSP3-7.0 (global radiative forcing of 7.0 Wm<sup>-2</sup> by 2100; hereafter S3) except that the LULCC input is taken from SSP1-2.6.

These LUMIP simulations cover the period from 2015 to 2100. The differences S3L1 minus S3 as well as S1 minus S1L3 allow an analysis of the biogeophysical climate response to afforestation and avoided deforestation in a medium- and low-warming world, respectively (Table 1).

Both simulation differences are based on the projected LULCC difference from S3 to S1, which shows reduced tropical deforestation and large-scale afforestation (scenario difference in global forest area is around 1000 Mha by 2100 (Lawrence et al., 2016)), and reduced expansion of cropland and grassland. Regions of avoided deforestation versus afforestation are shown in Fig. S1. Here, we focus on the average of the two scenario differences averaged over the last 50 years of simulation (i.e., 2051–2100) which we refer to as AFFOREST, i.e., AFFOREST =  $\frac{1}{2}$ (S1-S1L3) +  $\frac{1}{2}$ (S3L1-S3). Note that AFFOREST includes both avoided deforestation and afforestation as discussed above and shown in Fig. S1, though for brevity we will sometimes refer to it as an afforestation scenario in the text. Similarly, we define the average of the simulations with S3 LULCC as BASE, i.e., BASE =  $\frac{1}{2}$ (S1L3 + S3). Note that we focus on a 50-year timeperiod (compared to the more typical 30-year average) and the average across scenarios, which helps to increase the signal-to-noise ratio. We did not find a strong dependence of the climate response to afforestation on the low- to medium-warm background climate (see Section 3.1).

We selected seven Earth System Models (ESMs) from the models that performed these simulations based on the availability of multiple S1L3 and S3L1 ensemble members (ACCESS-ESM1-5, CESM2 and UKESM1-0-LL) or otherwise the availability

Table 1. Summary of LUMIP experiments (i.e., difference of two simulations) analysed in this study.

| Experiments                                      | Baseline climate | Interpretation                                                |  |  |
|--------------------------------------------------|------------------|---------------------------------------------------------------|--|--|
| SSP126 minus SSP126-SSP370Lu<br>(called S1-S1L3) | SSP1-2.6         | Afforestation + avoided deforestation in low warming world    |  |  |
| SSP370-SSP126Lu minus SSP370<br>(called S3L1-S3) | SSP3-7.0         | Afforestation + avoided deforestation in medium warming world |  |  |

of at least five S1 and S3 ensemble members (CNRM-ESM2-1, IPSL-CM6A-LR, MIROC-ES2L and MPI-ESM1-2-LR) to increase the signal-to-noise ratio. We had to exclude the NorESM2-LM model due to an unrealistic global mean temperature of up to 19.5°C in the S1L3 simulation and the CanESM5 model due to a different distribution of forest cover changes (including minimal forest cover change in the tropics) compared to other ESMs. We calculated the multi-model ensemble mean (MMM) from the ensemble means of these seven ESMs interpolated to a common  $2.5 \times 2.5^{\circ}$  grid. Further details about the resolution and land modelling schemes of the chosen ESMs are given in Table 2.

130

Statistical significance of changes was tested by propagating the uncertainty due to internal variability through this ensemble and scenario averaging (not shown), but a simpler approach based on where 6 of 7 models agree on the sign of change gave similar answers and is used throughout the text.

**Table 2.** List of CMIP6 models used that performed the LUMIP simulations. The number of ensemble members, horizontal resolution and details of the land modelling scheme are given. PFT stands for plant function type. Dynamic vegetation cover in climate models refers to the simulation of vegetation changes over time in response to climate conditions, CO<sub>2</sub> levels, and disturbances, allowing for feedback between ecosystems and the atmosphere. Information about the number of PFTs and dynamic vegetation cover are taken from Arora et al. (2020).

| Model name    | Ensemble members S1L3 & S3L1 | Ensemble members S1 & S3 | Horizontal resolution                 | No. of<br>PFTs | Dynamic vegetation cover | Irrigation | Citation<br>LUMIP<br>simulations |
|---------------|------------------------------|--------------------------|---------------------------------------|----------------|--------------------------|------------|----------------------------------|
| ACCESS-ESM1-5 | 10                           | 10                       | 1.875°×1.25°                          | 13             | No                       | No         | Ziehn et al. (2021)              |
| CESM2         | 3                            | 3                        | $1.25^{\circ} \times 0.9^{\circ}$     | 22             | No                       | Yes        | Danabasoglu (2019)               |
| CNRM-ESM2-1   | 1                            | 5                        | $1.4^{\circ} \times 1.4^{\circ}$      | 16             | No                       | No         | Seferian (2019)                  |
| IPSL-CM6A-LR  | 1                            | 5                        | $2.5^{\circ} \times 1.3^{\circ}$      | 15             | No                       | No         | Boucher et al. (2019)            |
| MIROC-ES2L    | 1                            | 5                        | $2.81^{\circ} \times 2.81^{\circ}$    | 13             | No                       | No         | Hajima et al. (2019)             |
| MPI-ESM1-2-LR | 1                            | 5                        | $1.8^{\circ} \times 1.8^{\circ}$      | 13             | Yes                      | No         | Pongratz et al. (2019)           |
| UKESM1-0-LL   | 5                            | 5                        | $1.875^{\circ}\!\times\!1.25^{\circ}$ | 13             | Yes                      | No         | Wiltshire et al. (2019)          |

The MMM difference in forest cover between 2051 and 2100 in AFFOREST is shown in Figure 1a. Afforestation and avoided deforestation are primarily located in central Africa, followed by North and South America, India, and central Europe. Conversely, deforestation is located in eastern China and eastern Europe. Zonal-mean changes reveal that the largest forest cover increase is located in the tropics, with a smaller increase in the boreal Northern Hemisphere (Fig. 1b). While some model differences regarding the forest cover changes exist (e.g., ACCESS-ESM1-5 shows a smaller latitudinal extent of African forest changes than other models), these differences stem mainly from variations in baseline forest cover (not shown). Decreases in crop- and grassland largely match the expansion in forest area, although the changes in grassland show notable differences across models (Fig. S2a-d). Out of the three models providing shrubland cover data, only ACCESS-ESM1-5 shows large expansions in the northern and southern boreal zones (Fig. S2e, f). Note that these simulations do contain other LULCC than forest and associated crop/grass/shrub changes such as changes in irrigation and wood harvest, although only CESM2 includes

the former and the latter can be related to deforestation changes. Further description of the LUMIP experiments can be found in Lawrence et al. (2016).

**Figure 1.** (a) Annual mean spatial pattern of tree fraction changes from 2051–2100 for AFFOREST. Stippling shows where 5 out of 6 ESMs agree on the sign of change. (b) Zonal-mean tree fraction change is shown for 6 ESMs and the MMM (black line). Note that MIROC-ES2L does not provide tree fraction data.

## 2.2 Moisture budget decomposition and related diagnostics

We apply a moisture budget decomposition, adapting the method from Seager et al. (2010) and Wills and Schneider (2016), to analyse the physical mechanism driving changes in tropical moisture fluxes and to disentangle the influence from circulation and transient eddies. Changes in the moisture budget are described by

$$\Delta(P - E) \approx \Delta MC + \Delta TE \tag{1}$$

where  $\Delta$  denotes the time-mean AFFOREST difference and  $\Delta MC$  and  $\Delta TE$  the contributions of time-mean circulations and transient eddies to changes in moisture-flux convergence, respectively. Precipitation P and evapotranspiration E are derived from model output. The mean-circulation moisture-flux term  $\Delta MC$  can be split into dynamic and thermodynamic components as

$$\Delta MC = -\nabla \cdot \left\langle (\Delta \mathbf{u}) q_{Base} \right\rangle - \nabla \cdot \left\langle \mathbf{u}_{Base}(\Delta q) \right\rangle$$
(2)

where  $\nabla$  is the nabla operator on a sphere, **u** is the horizontal wind vector, q is the specific humidity and  $\langle \cdot \rangle$  indicates an integration from the pressure at the top of the atmosphere to the surface normalised by the gravitational acceleration such that

the units are the same as for precipitation. The dynamic contribution  $\Delta MC_{Dyn}$  arises only from changes in mean atmospheric circulation and the thermodynamic contribution  $\Delta MC_{Thermo}$  arises only from changes in specific humidity (Eq. 2).

The transient eddy term  $\Delta TE$  in Eq. (1) captures changes in concurrent sub-seasonal and higher frequency variations in specific humidity and circulation. This term is calculated as the residual from Eq. (1), as daily data is not available for all ESMs to allow a direct calculation of this term, and it therefore also includes the small nonlinear mean-circulation term, which arises from the product of changes in humidity and circulation.

#### 2.2.1 Surface and near-surface impact on moisture flux





Insights into the impact of near-surface circulation changes on  $\Delta(P-E)$  in the (sub)tropics can be gained from the omega scaling from Wills and Schneider (2015). This omega scaling is an empirical approximation of  $\Delta MC$  based on the integral form of the mean value theorem and is given by

$$\Delta(P-E) \approx -\frac{1}{a}\Delta(q_s\omega_{700})\tag{3}$$

with near surface-specific humidity  $q_s$  (at 2m height) and 700-hPa vertical pressure velocity  $\omega_{700}$ . Note that the formula includes  $\omega$  at 700 hPa, rather than 850 hPa as in Wills and Schneider (2015) and Wills and Schneider (2016), which was empirically found to better match net precipitation in our analysis and is consistent with the higher lifting condensation level (LCL) over tropical land rather than ocean (see Sec. 3.2.2.). Nonetheless, this approximation gives only a qualitative description of  $\Delta(P-E)$  since it neglects the effect of transient eddies and horizontal advection, and is not defined in regions where the surface extends above 700 hPa.

One important influence on  $\omega_{700}$  and thus  $\Delta(P-E)$  is the orographic effect of moisture flow across surface pressure gradients (i.e., upward movement on the windward side and downward movement on the leeward side of orography). The surface contribution  $\Delta S$  can be calculated as

$$\Delta S = -\frac{1}{g}\Delta(q_s\omega_s) = -\frac{1}{g}\Delta(q_s\mathbf{u}_s \cdot \nabla p_s) \tag{4}$$

where  $p_s$  denotes surface pressure,  $\omega_s$  denotes near-surface vertical pressure velocity and  $u_s$  denotes near-surface winds. In our results, the change in  $\Delta S$  primarily reflects changes in aerodynamic drag on the near-surface wind  $(u_s)$  caused by the alteration of surface roughness. Note that near-surface temperature and humidity variables are defined at a height of 2 m, while near-surface wind variables are defined at 10 m, consistent with CMORized CMIP6 variables.

#### 2.2.2 Net energy input impact on moisture flux

We also analyse changes in the net energy input (NEI) to investigate how energetic changes influence atmospheric circulation and net precipitation. In steady state, the moist static energy equation relates NEI to the divergence of moist static energy flux:

$$NEI = \nabla \cdot \langle \mathbf{u}h \rangle. \tag{5}$$

Here, moist static energy (MSE) is given by  $h = c_p T + L_v q + \Phi$ , with the specific heat capacity of air at constant pressure  $c_p$ , temperature T, latent heat of vaporization  $L_v$ , three-dimensional specific humidity q, and geopotential  $\Phi$ . In the tropics,

changes in MSE flux divergence are related to moisture flux convergence and P - E, because energy is transported in the opposite direction as moisture in tropical overturning circulations, leading to anomalous P - E in regions of anomalous NEI. NEI can be further separated into individual components due to the surface latent heat flux (LHF), surface sensible heat flux (SHF), net radiative energy flux into the atmospheric column ( $R_{NET}$ ; comprising both short- and long-wave contributions at the top of the atmosphere and surface) and the atmospheric energy storage (ST):

$$NEI = LHF + SHF + R_{NET} + ST$$
(6)

Here, we use the convention that positive values represent fluxes into the atmospheric column. The values for LHF, SHF,  $R_{NET}$  and ST can be calculated from the LUMIP data. ST is found to be negligible and is not considered in the later analysis (not shown).

#### 3 Results







#### 3.1 Climate response over land

Here, we analyse the climate response over land to the future afforestation scenario (including avoided deforestation, as discussed in Section 2.1). In response to the afforestation, the zonal-mean evapotranspiration increases in the tropics due to enhanced transpiration through a larger leaf area, increased evaporative capacity due to deeper root systems and increased surface roughness (Fig. 2b; Bonan, 2008). The increase in evapotranspiration enhances the atmospheric moisture content, leading to an increase in precipitation over the ITCZ region over land throughout the year, while precipitation decreases north of the equator during the extended Northern Hemisphere monsoon period (i.e., March to September (MJJAS)) (Fig. 2a). The surface temperature decreases due to the large increase in cooling by evapotranspiration, which dominates over the warming due to the land surface albedo increase in the tropics (Fig. 2d). Notably, these changes in the hydroclimate and temperature show large agreement across models in the latitudinal structure and seasonality. Additionally, these tropical climate responses are largely independent of the low- (S1-S1L3) and medium- (S3L1-S3) warming background climate, as shown by analysing the simulations separately (Fig. S3).

The net precipitation (P-E) quantifies the exchange of moisture between the atmosphere and Earth's surface and is a key control of surface runoff. In response to afforestation, net precipitation shows a predominant decrease north of the equator throughout the year and south of the equator in MJJAS, accompanied by a modest increase along the ITCZ band (Fig. 2c). From a land-climate perspective, this tropical  $\Delta(P-E)$  response could be seen as an increase in moisture recycling, since it means that precipitation increases less than evapotranspiration and thus more of the moisture is sourced locally. However, this response seems counterintuitive from an atmospheric dynamics perspective: Despite a simulated increase in NEI (given by the sum of the surface latent heat, surface sensible heat and net radiative energy into the atmospheric column; Eq. 6) over the tropics due to afforestation (Fig. 2e), we observe an overall reduction in net precipitation. Based on atmospheric dynamics, we would expect that an increase in tropical NEI implies a strengthening of the overturning circulation and an increase in net precipitation throughout the inner tropics. While the southward shift and contraction towards the equator of the ITCZ are

Figure 2. Zonal-mean monthly changes in (a, f) precipitation  $\Delta P$ , (b, g) evapotranspiration  $\Delta E$ , (c, h) net precipitation  $\Delta (P-E)$ , (d, i) Surface temperature  $\Delta Ts$ , (e, j) net energy input  $\Delta NEI$  over global land (upper row) and over Africa (lower row; average over land area  $20^{\circ}$ W to  $50^{\circ}$ E) in AFFOREST. Contours show the BASE values with a spacing of 1.5 mm day<sup>-1</sup> for  $\Delta P$ ,  $\Delta E$  and  $\Delta (P-E)$ , a spacing of  $5^{\circ}$ C for  $\Delta Ts$  and of 20 W m<sup>-2</sup> for  $\Delta NEI$ . Dashed contours indicate negative values. Stippling shows where 6 out of 7 ESMs agree on the sign of change.

consistent with energetic frameworks based on NEI (Bischoff and Schneider, 2014; Byrne and Schneider, 2016), at least for the case of positive gross moist stability, the overall reduction of P - E is not.

The increase in NEI is mainly determined by the increase in latent heat flux, partially compensated for by a decrease in net radiative energy into the atmosphere and a minimal change in sensible heat flux (Fig. S4). The net radiative changes result from changes in longwave radiation at the top of the atmosphere (ToA) and surface rather than changes in shortwave radiation (not shown). Overall, this indicates a stronger greenhouse effect over the forest, where increased water vapor and cloud cover trap outgoing radiation. As a result, the surface cools less efficiently (Fig. S4f) and less longwave radiation escapes at the top of the atmosphere (Fig. S4e). The opposing effects in ToA and surface longwave radiation combine with latent heat flux to give the net effect in NEI.



This study thus seeks to identify the mechanisms driving the  $\Delta(P-E)$  pattern over the tropics from a dynamics perspective and to reconcile why the increase in NEI does not lead to the dynamically expected increase in (P - E). Later analysis focuses on the response over Africa which we identify to be the dominant region influencing the hydroclimate, temperature, and NEI response over the tropics (Fig. 2f-j, S5). This finding aligns with projected differences in forest cover between S1 and S3, which are most pronounced over Africa (Fig. 1a). Note that while the Amazon also shows consistent hydrological and temperature

responses, it is smaller in magnitude, consistent with the less extensive forest cover difference between the S1 and S3 scenarios (Fig. 1, S1).

#### 225 3.2 Mechanistic insights into moisture flux changes over Africa

## 3.2.1 Moisture budget analysis





The following analysis focuses on hydroclimate changes over Africa and the underlying mechanisms. The analysis is split up into the two extended monsoonal seasons influencing the monsoon precipitation over western Africa (May to September, MJJAS) and southeastern Africa (November to March, NDJFM) (Douville et al., 2021). The spatial pattern of precipitation change shows an increase in equatorial Africa in MJJAS, while precipitation decreases over large parts of the western African monsoon region (Fig. 3a; cf. climatology in Fig. S6a). During NDJFM, precipitation increases over the southern African monsoon region (Fig. 4a; cf. climatology in Fig. S6e). Evapotranspiration increases in both seasons over central Africa, the region of largest afforestation and avoided deforestation between the SSPs (Fig. 3b, 4b). This leads to a tripole dry-wet-dry pattern in net precipitation over Africa in MJJAS and NDJFM, with varying extents of the wettening (Fig. 3c, 4c). Thus, the climatological dry P - E region over southeastern Africa and wet region over central Africa are strengthened in MJJAS (Fig. 3c; cf. climatology in Fig. S6c), as are the climatological wet P - E region over southeastern Africa and dry region over western Africa in NDJFM (Fig. 4c; cf. climatology in Fig. S6g). Note that the non-zero changes in P - E imply that the afforestation in this region leads to changes in atmospheric circulation and moisture transport.

We perform a moisture budget decomposition as described in Section 2.2 to investigate the underlying dynamical and thermodynamical changes, and associated mechanisms leading to the  $\Delta(P-E)$  pattern. The net precipitation changes in both monsoonal seasons are primarily driven by changes in the mean-circulation moisture flux  $\Delta MC$  (Fig. 3d, 4d). There is relatively large agreement among the models regarding the magnitude and spatial pattern of the  $\Delta MC$  term, albeit with some spatial variability across individual models (Fig. S7, S8). Nonetheless, in all models, the  $\Delta MC$  term emerges as the dominant contributor to the  $\Delta(P-E)$  response during both seasons (Fig. S7, S8). In particular, the dynamic component contributes to this  $\Delta MC$  pattern (Fig. S9b,e), while the thermodynamic component is small despite large and robust changes in specific humidity in central Africa (Fig. S9c, f; cf. specific humidity changes in Fig. S10a, e). The transient-eddy changes are generally smaller in magnitude than the mean-circulation-related changes and do not primarily determine the overall pattern across Africa. However, changes can be locally important, such as the contribution to the net precipitation drying in southeastern Africa in MJJAS (Fig. 3e) and wettening over central Africa in NDJFM (Fig. 4e). Furthermore, in specific regions like western equatorial Africa, large and opposing  $\Delta TE$  and  $\Delta MC$  anomalies can result in near-zero net change (Fig. 3c-e), highlighting local compensations. Nonetheless, the primary mechanism(s) determining the structure of the overall pattern in  $\Delta(P-E)$  over Africa are of dynamic origin (i.e., related to changes in the time-mean circulation), rather than related to transient-eddy or thermodynamic changes.

Figure 3. Spatial maps of MJJAS changes in (a) precipitation  $\Delta P$ , (b) evapotranspiration  $\Delta E$ , (c) net precipitation  $\Delta (P-E)$ , (d) mean-circulation moisture flux term  $\Delta MC$ , (e) transient eddy term  $\Delta TE$ , (f) omega scaling and (g) surface term  $\Delta S$  over Africa in AFFOREST. Pink contours enclose the regions where there is at least 5% of afforestation. Stippling shows where 6 out of 7 ESMs agree on the sign of change. Arrows show the relationship between the different variables.

Figure 4. Same as Figure 3 but for NDJFM.

### 3.2.2 Surface and lower-tropospheric changes




Insights into the impact of near-surface changes on ΔMC and thus the Δ(P – E) pattern can be gained from the empirical omega scaling (Eq. 3). The omega scaling shows overall agreement with the net precipitation changes (Fig. 3f, 4f). Note that the formula includes ω at 700 hPa, in contrast to 850 hPa as in Wills and Schneider (2015) and Wills and Schneider (2016), which we found to better match net precipitation in our analysis and is consistent with the higher LCL over tropical land rather than ocean (Fig. S11). Additionally, while the formula for the omega scaling incorporates specific humidity changes (Eq. 3),
we found the contribution of the thermodynamic effects to Δ(P – E) to be minimal. This emphasizes the dominant impact of dynamical changes in convergence below 700 hPa on Δ(P – E).

An important contributor to  $\omega_{700}$  and thus  $\Delta(P-E)$  is the orographic effect of moisture flow across surface pressure gradients, which we quantify with the surface term  $\Delta S$  (Eq. 4). Following afforestation,  $\Delta S$  shows a consistent decrease near the mountain ridges along the southeast, southwest, and west coasts of Africa during both seasons (Fig. 3g, 4g; cf. climatology in Fig. S6d, h). This response can be attributed to an afforestation-induced increase in surface drag, which slows down moisture-laden near-surface winds originating from the ocean (Fig. 5a, c). The wind speed reduction is around 8% averaged across Africa (Fig. 5b, d) and it reaches more than 15% in central Africa (Fig. S12a, c). This is accompanied by a concurrent increase in wind stress of a similar magnitude (Fig. S12b, d), indicating that the wind speed reduction results from an increase in roughness length due to afforestation. This low-level flow reduction is further corroborated by a significant weakening of the inland low in sea-level pressure (Fig. S10b, c).

The reduced wind speed hinders moisture transport and diminishes upward motion along the topographic slopes by reducing the near-surface vertical wind component  $\omega_s \equiv \mathbf{u}_s \cdot \nabla p_s$  (shown by the pink contours in Fig. 5a, c). This contributes to a reduction in vertical motion at the LCL, which reduces topographically-induced net precipitation and results in a negative signal in  $\Delta S$ . Note that this signal is not directly visible in the precipitation pattern (Fig. 3a, 4a) due to the modification of vertical velocities between the surface and the top of the LCL, as described in the following section. This surface effect extends throughout the lower troposphere, with wind speed reductions over Africa falling off approximately logarithmically until around 700 hPa (Fig. 5b, d). Note that we interpolated the pressure vertical velocity profiles to sigma coordinates (defined as pressure normalized by the grid-cell surface pressure in the BASE dataset). This coordinate transformation was performed to provide a clearer representation of effects on surface values, as pressure coordinate surfaces vary in their height above the surface, while in sigma coordinates  $\sigma = 1$  always corresponds to the surface itself. The resulting sigma coordinates are presented in units of pressure by multiplying by the area-averaged surface pressure from the BASE dataset. Overall, this surface-drag induced slow-down of lower-tropospheric winds and associated vertical pressure velocity have a substantial influence on  $\Delta(P-E)$ , although other effects contribute to the observed discrepancies between  $\Delta S$  and  $\Delta (P-E)$ . The overall decrease in net precipitation occurs despite a thermodynamically more favorable environment for precipitation, marked by an increase in near-surface specific humidity and MSE (Fig. S10). The influence of the thermodynamic environment will be investigated with an atmospheric energy budget perspective in the next section, but the fact that the net precipitation decrease still occurs suggests the importance of the mechanical weakening in opposing the influence of the latent heating and near-surface MSE changes.

Figure 5. (a, c) Spatial maps of changes in near-surface wind speed (filled contours) over Africa for AFFOREST in MJJAS (upper row) and NDJFM (lower row). Pink contours show changes in near-surface pressure vertical velocity  $\omega_s$  with solid (dashed) contours indicating positive (negative) values. Vectors show BASE surface winds. (b, d) Vertical profile of percentage changes in wind speed over Africa (land region of  $37^{\circ}\text{S}-15^{\circ}\text{N}$ ,  $20^{\circ}\text{W}-50^{\circ}\text{E}$ ) interpolated to sigma coordinates. The sigma coordinates are calculated as pressure divided by the grid-cell surface pressure in BASE and multiplied by the area-averaged surface pressure in BASE to get sigma coordinates in pressure units. The envelope shows the  $25^{\text{th}}$  to  $75^{\text{th}}$  percentile model spread. The light blue dot shows the percentage changes in surface wind speed (positioned at the average surface pressure in BASE) with the error bar showing the  $25^{\text{th}}-75^{\text{th}}$  percentile model spread.

# 3.2.3 Free-tropospheric changes

We next investigate free-tropospheric mechanisms influencing  $\Delta(P-E)$  as well as the reasons for the apparent discrepancy with NEI changes. For this, we examine spatial patterns of NEI over Africa and pressure vertical velocity profiles over the alternating dry-wet-dry regions of  $\Delta(P-E)$  in AFFOREST (Fig. 6, 7). Note that the same conclusions are reached when focusing on near-surface moist static energy instead of NEI (Fig. S10d, h).

Over western and southeastern Africa, P-E decreases in MJJAS and NDJFM (Fig. 6a, 7a). The afforestation-induced changes in pressure vertical velocity (light blue lines in Fig. 6c,e; 7c,e) oppose the climatological motion (dark blue line in Fig. 6c,e; 7c,e). This leads to a weakening of the upwards motion over these regions and thus a negative  $\Delta(P-E)$  response. The western Africa response in NDJFM and the southeastern Africa response in MJJAS show pressure vertical velocity anomalies concentrated in the lower troposphere, indicating that the surface drag-induced change is the dominant effect impacting the pressure vertical velocity profile over western and southeastern Africa in these seasons. The MJJAS response over western Africa shows small changes in the lower-troposphere, albeit with large spatial differences within the region (Fig. 3f), indicating the P-E response in this region results from a complex interplay of different mechanisms, e.g., including the influence of a strengthened Saharan heat low (Fig. S10b, c).

Conversely, over central Africa, P-E increases in MJJAS and NDJFM (Fig. 6a, 7a). This response is related to the strong increase in NEI over central Africa, which is dominated by an increase in latent heat flux and the associated longwave radiative effects in a moister and cloudier atmosphere (Fig. 6b, 7b, S4). This excess NEI strengthens the climatological upward motion in the free troposphere (i.e., AFFOREST changes in the direction of BASE values in Fig. 6d and 7d) by enhancing convection. Consequently, the climatological wet P-E region over central Africa is strengthened (Fig. 6d, 7d; cf. climatology in S6c, d). During NDJFM, the increase in NEI is overall smaller than in MJJAS but the NEI maxima over Africa are located closer to the deep convective region (i.e., region of largest climatological NEI), leading to a larger region of  $\Delta(P-E)$  wettening. Note that near-surface pressure vertical velocity is weakened over central Africa due to the surface drag-induced effect described previously.

These findings reveal that net precipitation changes over Africa are driven by the complex interplay of afforestation effects on lower tropospheric (until around 700 hPa) circulation changes, through changes in surface drag and through the influence of changes in NEI on deep convection. While increased surface drag reduces P-E in orographic regions by slowing down near-surface winds and associated lower-tropospheric vertical velocities, the increased NEI increases P-E by strengthening convective upper-tropospheric circulations. The regional sign of  $\Delta(P-E)$  then depends on the dominance of the surface drag or energetic effect over different African subregions. These findings help to reconcile the NEI pattern with the  $\Delta(P-E)$  response. Another way of thinking about how our results reconcile the NEI and P-E change is to consider that the idea that positive NEI anomalies lead to positive net precipitation anomalies assumes pressure vertical velocity changes to be the same sign throughout the atmospheric column, which is not true in AFFOREST due to differing drivers of change in the upper and lower troposphere. The changes in the vertical profile of pressure vertical velocity correspond to a change in the gross moist stability, which determines how deep-convective circulations respond to NEI.

## 3.2.4 Summary of mechanisms governing net precipitation response







Here, we summarise the surface drag and energetic mechanisms we determined to influence moisture fluxes and P-E following afforestation (Figure 8). In a base state without trees, evapotranspiration is relatively low over crop- and grassland (Figure 8a). Orographic precipitation occurs as moisture-laden winds ascend topographic slopes, reach the LCL and condense into clouds. Convective precipitation, on the other hand, is closely linked to surface evapotranspiration, which increases the

Figure 6. Spatial maps of MJJAS changes in (a) net precipitation  $\Delta(P-E)$  and (b) net energy input  $\Delta NEI$  over Africa in AFFOREST. In (b), contours show BASE values of NEI with a spacing of 20 W m<sup>-2</sup>. (c-e) Vertical profile of BASE values (dark blue) and changes following AFFOREST (light blue) in pressure vertical velocity  $\omega$  averaged over the western, central and southeastern African regions indicated in (a). The dark blue dot shows the BASE value and light blue one the AFFOREST value of the near-surface pressure vertical  $\omega_s$  with error bars showing the 25-75<sup>th</sup> percentile. Note that BASE values have been divided by 50 to display them on the same axes and the profiles are interpolated to sigma coordinates. The sigma coordinates are calculated as pressure divided by the grid-cell surface pressure in BASE and multiplied by the area-averaged surface pressure in BASE to get sigma coordinates in pressure units. The surface values (dots) are positioned at the average surface pressure in BASE over the relevant region.

propensity for convection and associated precipitation. The resulting net precipitation is positive in most regions, leading to runoff and water availability. Following afforestation, two competing mechanisms shape the hydroclimate response over Africa. The first is the surface drag effect, in which increased surface drag following afforestation slows moist onshore winds, in turn reducing near-surface and lower-tropospheric pressure vertical velocity (Figure 8b). This leads to a suppression of orographic

**Figure 7.** Same as Figure 6 but for NDJFM. Also note that the regions marked as western, central and southeastern Africa are not the same as in Fig. 6.

precipitation in response to afforestation, such that precipitation increases less than evapotranspiration, leading to a decrease in net precipitation compared to the base state. The second mechanism is the energetic effect (Figure 8c), where afforestation increases the net energy input to the atmosphere due to greater latent heat flux (NEI<sub>LHF</sub>), partially offset by increased radiative cooling of the atmosphere (NEI<sub>Rnet</sub>). This leads to a strengthening of the climatological vertical upward motion in the free troposphere. The associated enhancement of convective precipitation results in an increase in net precipitation compared to the base state. Our results indicate that net precipitation changes over Africa in response to afforestation are driven by a complex interplay between lower- and upper-tropospheric circulation changes. Both the energetic and the surface-drag effect are active over the entire African domain to afforestation, however the resulting net precipitation response depends on which effect is dominant (i.e., has the larger influence on vertical velocity at 700 hPa and thus P - E) over a certain region. The surface drag effect is the dominant driver of the net precipitation decrease over western and southeastern Africa during MJJAS and NDJFM,


although the magnitude of change is modulated by the energetic effect. Conversely, the energetic effect is the dominant driver of the net precipitation increase over central Africa, though the surface drag effect still influences near-surface pressure vertical velocity. Overall, this dynamic interaction between local surface processes and large-scale atmospheric energy balances explains the spatial patterns of hydroclimate change seen in our study.

Figure 8. Schematic illustrating (a) the mechanisms affecting the hydroclimate in a base state without trees, along with the impacts of (b) surface drag and (c) energetic changes on moisture flux resulting from afforestation. Moisture fluxes (P, E, P - E) are represented by dark blue arrows, pointing in the direction of flow. In (b) and (c) the climatological moisture and energetic fluxes from (a) are depicted as light blue arrows for reference. The wind flow over topography is shown as black arrows (labelled  $\mathbf{u}$ ) and vertical winds are depicted by upward-pointing black arrows (labelled  $\omega$ ). Convective fluxes are represented by curvy grey arrows, with outflow indicated by curved grey arrows at the top of the cloud. Stronger fluxes and circulation are represented by thicker and longer arrows, while weaker fluxes are indicated by thinner and shorter arrows. Similarly, stronger precipitation intensity is shown by larger raindrops and cloud symbols and vice versa for lower precipitation intensity. The lifting condensation level is labelled as LCL, orographic precipitation as  $P_{orog}$  and convective precipitation as  $P_{conv}$ . NEI<sub>LHF</sub> and NEI<sub>Rnet</sub> refer to the latent heat flux and the net radiative energy component of the net energy input NEI, respectively.

#### 345 4 Discussion and conclusions

In this study, we analysed hydroclimate changes in the tropics under future afforestation scenarios using simulations from the Land Use Model Intercomparison Project (LUMIP; Lawrence et al., 2016). Our primary focus was on hydroclimate and atmospheric changes over Africa, which undergoes significant hydrological changes in these simulations but remains understudied compared to the Amazon rainforest. We identify three key findings:

- 1. Afforestation and avoided deforestation lead to robust increases in precipitation and evapotranspiration, alongside widespread decreases in net precipitation (precipitation minus evaporation) in the tropics, particularly over Africa.
  - 2. The tropical hydroclimate response to future forest cover change is largely independent of the background climate under low- and medium-warming scenarios.
- The changes in net precipitation over Africa are driven by the competing effects of surface drag-induced reduction of lower-tropospheric winds and net energy input-induced strengthening of deep-convective upper-tropospheric circulations.

Here, we compare these findings to existing literature to contextualize our results.

Our results show that the hydroclimate response is most pronounced in the regions with the greatest forest cover changes, namely over Africa (Fig. 1, 2). The African climate response dominates the overall tropical hydroclimate response over land, highlighting the larger regional versus remote responses to tropical forest cover changes, although net precipitation anomalies provide evidence of changes in moisture transport by the large-scale circulation. This aligns with the findings of Devaraju et al. (2015) who demonstrated that while large-scale deforestation in the northern mid- and high latitudes can influence the ITCZ remotely, tropical forest cover changes primarily induce regional hydroclimate responses. Across models, we find a robust increase in evapotranspiration by 5–10% following afforestation, which can be attributed to enhanced leaf area and deeper root systems (Bonan, 2008), as well as larger turbulent fluxes due to surface roughness changes (Burakowski et al., 2018; Sud et al., 1988). Our results further show a consistent increase in precipitation (by up to 5% in the MEM) in response to afforestation which is in agreement with satellite-based studies reporting significant local increases in precipitation to recent increases in global vegetation cover (Zeng et al., 2018) and vice versa for tropical deforestation (Smith et al., 2023). These findings are further supported by previous single model modelling studies demonstrating that deforestation in Africa leads to a local reduction in precipitation (e.g. Varejao-Silva et al., 1998; Semazzi and Song, 2001; Nogherotto et al., 2013). However, multimodel studies — for instance investigating the precipitation response to large-scale idealised deforestation — find that models do not show consistent precipitation changes in many regions (Luo et al., 2022; Boysen et al., 2020). Thus, the consistent precipitation (and net precipitation) changes over Africa across multiple models in response to future forest cover changes represents a new insight. Taken together, these findings emphasize the critical role of regional forest cover in modulating regional hydroclimate conditions.

We find that the temperature and hydroclimate responses to afforestation in the tropics are largely independent of the background climate under low- to medium-warming scenarios (Fig. S3). Previous studies have highlighted that the biogeophysical

cooling effect of vegetation is more significant in a warming world (Pitman et al., 2011; Hua and Chen, 2013), although these studies primarily concentrated on the snow-albedo feedback of LULCC in mid- and high-latitudes which weakens under increased CO<sub>2</sub> emissions. Studies incorporating both biogeophysical and biogeochemical effects suggest that the cooling effect of an increase in global vegetation cover is enhanced under high-emission scenarios due to the CO<sub>2</sub> fertilisation effect (Alkama et al., 2022; Kim et al., 2017), with around 20% of the total cooling attributable to biogeophysical processes (Alkama et al., 2022). However, when compared to overall global warming in a fractional sense, the cooling potential of LULCC declines in high-emission scenarios (Alkama et al., 2022; Loughran et al., 2023). Despite these insights on the background-climate dependence of temperatures responses, few studies have specifically addressed how hydroclimate responses to LULCC depend on the background climate, particularly in the tropics. To the best of our knowledge, this is the first study that documents the mean-state independence of the afforestation-induced hydroclimate response under low-to-medium climate warming.

The mechanisms underlying precipitation changes due to forest cover alterations have been explored by some previous studies (Khanna and Medvigy, 2014; Crook et al., 2023; Nogherotto et al., 2013; Abiodun et al., 2008; Lejeune et al., 2015; Sud et al., 1988, 1996; Cheng and Mccoll, 2023; Bell et al., 2015; Luo et al., 2022; Arnault et al., 2023; Akkermans et al., 2014). Research on Amazonian deforestation suggests that small-scale or patchy deforestation increases sensible heat flux while reducing latent heat flux, resulting in surface warming and mesoscale convection that can enhance local precipitation (Baidya Roy and Avissar, 2002; Khanna and Medvigy, 2014). A recent study by Qin et al. (2025) found that different mechanisms impact rainfall during the dry and wet season after Amazonian deforestation. During the wet season, they found that deforestation warms the land surface, creating a low pressure area that draws in moisture and increases precipitation over deforested patches. During the dry season, they found that deforestation reduces evapotranspiration and lower atmospheric moisture levels leading to a decrease in precipitation across a wider region. However, these previous studies largely lack a more quantitative analysis of the underlying mechanisms through atmospheric dynamics methods. Additionally, these studies all consider precipitation, which is directly influenced by the evapotranspiration changes associated with forest cover changes, and our results indicate that net precipitation changes (associated with moisture flux changes) often change in the opposite sense as precipitation, indicating the need to consider net precipitation in studies of the hydroclimate response to forest cover change.

The impact of forest cover changes on net precipitation has been investigated by only a few studies, including some using atmospheric moisture tracking models (Zan et al., 2024; Cui et al., 2022) and some using regional (Lejeune et al., 2015) or global climate models (Luo et al., 2022). Previous studies using moisture tracking models, which rely on climatological circulation patterns, do not account for large-scale circulation adjustments. However, our findings indicate that neglecting these circulation adjustments leads to an incomplete understanding of the hydroclimatic impacts of forest cover changes, as circulation responses play a leading-order role in determining net precipitation variations (Fig. 3, 4, S9). Similarly, Luo et al. (2022) found that net precipitation changes over Africa to an idealised deforestation scenario are primarily influenced by mean-circulation moisture flux based on a moisture budget analysis. While their findings agree with our results, they find less model agreement, which we speculate is due to the larger relative importance of extratropical afforestation in their simulations and fewer ensemble members used in their analysis.

Our results show that the surface roughness increase due to afforestation reduces net precipitation, which aligns with previous work focusing on roughness-induced changes in precipitation (Bell et al., 2015; Crook et al., 2023; Lejeune et al., 2015; Sud et al., 1988, 1996). These studies found that the reduction in surface roughness due to deforestation enhances lower-tropospheric wind speeds, which can enhance moisture transport from the ocean to deforested regions (Bell et al., 2015; Crook et al., 2023; Lejeune et al., 2015; Sud et al., 1988, 1996). For example, Bell et al. (2015) determined that complete deforestation of the Congo primarily influences regional precipitation via changes in surface roughness, which strengthens horizontal winds over the deforested area. The intensified winds enhance moisture transport from the Atlantic Ocean and western Congo, leading to increased moisture convergence west of the Rift Valley highlands and consequently greater rainfall over the eastern Congo Basin. Our results support the decrease in near-surface wind speed due to afforestation (Fig. 5, S12). However, in contrast to previous studies, we further link these surface drag-induced circulation changes to their impact on orographic precipitation and subsequent net precipitation changes (Fig. 3f, 4f), which has not been previously documented.








Our study demonstrates the competition between the surface drag effect found in previous studies (Bell et al., 2015; Crook et al., 2023; Lejeune et al., 2015; Sud et al., 1988, 1996) and the energetic effect, due to change in net energy input to the atmosphere, in setting net precipitation changes over Africa. In particular, we identify a novel climate dynamical mechanism where P-E changes occur opposite to both precipitation (P) and NEI in certain regions (i.e., southeastern and western Africa) due to the competition between the surface drag and energetic effects, while the NEI-driven changes dominate over climatologically ascending regions (i.e., central Africa) (Fig. 6, 7). Only one study by Lejeune et al. (2015) focusing on Amazonian deforestation has explicitly examined the role of convective processes alongside the influence of surface roughness on net precipitation response. They found a longitudinal precipitation dipole and increase in net precipitation over Amazonia to regional deforestation. They attributed the precipitation response to a combination of strengthened moisture-carrying winds due to decreased surface roughness, and reduced pressure vertical velocity and convection due to reduced net radiation at the surface and thus reduced latent heat flux. However, they acknowledge that the use of a regional climate model might dampen large-scale circulation changes and related precipitation changes, due to the imposed lateral boundary conditions. In our study focused on Africa in global simulations, we similarly find that surface roughness changes and convective changes are important (Fig. 8). Furthermore, we extend their previous work by applying robust methods like the moisture budget decomposition, linking these convective changes to large-scale energy budget modifications, and showing that a similar mechanism operates across tropical Africa. Overall, our findings underscore the necessity of considering both surface roughness and energetic mechanisms when evaluating the hydrological impacts of land cover changes.

Our work represents the first comprehensive analysis of the tropical hydroclimate impacts and underlying mechanisms associated with future "realistic" afforestation scenarios over Africa. Compared to previous studies, we were able to reduce the model and internal variability uncertainty in the results by using multiple ESMs and multiple ensemble members for each model. While our use of a large multi-model ensemble strengthens the robustness of our overall conclusions, the inter-model differences in the simulated hydroclimate and circulation responses also offer valuable insights into the uncertainties and complexities inherent in climate projections (Fig. S7, S8). Although some previous research has highlighted the importance of

surface roughness in influencing precipitation, we build on this to demonstrate the competing effects of surface roughness and energetic effects on net precipitation.






However, some uncertainties remain. Firstly, while we attribute the net precipitation changes to regional variations in forest cover over Africa, alterations in forest cover across other regions may induce non-local effects that modulate the hydroclimate response to regional forest cover over Africa. Secondly, there might be interactions between the identified surface drag and the energetic effects, since near-surface convergence, which we relate to surface drag changes, can also influence convection and modulate the energetic effect. Thirdly, differences remain in the mean circulation change patterns across models, although the primary drivers of net precipitation change remain the same across models. Fourthly, we were unable to perform single-perturbed parameter simulations, such as altering surface drag alone, to further validate our mechanistic analysis. Most previous modeling approaches modified surface roughness by adjusting vegetation height (e.g. Bell et al., 2015; Lague et al., 2019), which however simultaneously affects momentum, heat, and moisture fluxes (i.e., including evaporation). This interdependence makes it challenging to isolate the effects of surface drag alone. Instead, we propose running simulations where the roughness length for momentum is modified independently of the roughness lengths for heat and moisture. This simulation setup would allow the isolation of the pure surface (aerodynamic) drag effect on near-surface wind speed from the aerodynamic influences on evaporation and sensible heat flux. Finally, we focused solely on biogeophysical effects and there may be competing biogeochemical factors that influence the hydroclimate, which would require additional simulations incorporating both biogeophysical and biogeochemical effects.

Overall, our study contributes to an improved understanding of land-atmosphere interactions and associated hydroclimate changes in the tropics, particularly Africa. These findings underscore the importance of forest-circulation-climate feedbacks for accurately predicting hydroclimate responses to afforestation, with significant implications for sustainable forest and water resource management. Additionally, our finding that afforestation reduces net precipitation implies (with opposite sign) an increase in net precipitation and the potential for flooding in response to deforestation, which is crucial information for climate change adaption management. Given that most current afforestation strategies do not consider the potential hydroclimate effects, future scientific advances are essential to better assess the atmospheric and hydrological impacts of large-scale forest cover changes. This will ultimately help guide more effective afforestation strategies aimed at achieving net-zero emissions.

Data availability. The LUMIP simulations used in this study are published by the CMIP6 archive available via the Earth System Grid Federation (ESGF) at https://esgf-metagrid.cloud.dkrz.de/search/cmip6-dkrz/. The schematic was created in PowerPoint using vector images from Vecteezy.

Author contributions. All authors contributed to the conceptualization of this study. NLSF performed the data analysis and calculations,
 visualised the results, and wrote the initial draft. RCJW and SJDH contributed to the interpretation and discussion of the results, and edited the manuscript.

Competing interests. The contact author has declared that none of the authors has any competing interests.

## **Funding information**

This work was supported by an ETH Zurich Research Grant. NLSF and RCJW additionally acknowledge funding from the Swiss National Science Foundation (Award PCEFP2 203376). SJDH acknowledges funding by the Belgian Federal Science Policy Office BELSPO (B2/223/P1/DAMOCO and SR/00/410/AFROCARDS).

Acknowledgements. The authors would like to thank Felix Jäger, Peter Lawrence, and Marysa Laguë for the interesting and helpful discussion of some of the presented results. The authors would also like to thank Urs Beyerle for his help with the data downloading.

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
