# Peer review of "Mechanistic insights into tropical circulation and hydroclimate responses to future forest cover change"

_EGUsphere, 2025_

## Author Comment (AC1)

**Comments on "Mechanistic insights into tropical circulation and hydroclimate responses to future forest cover change" by Fahrenbach et al.**

The manuscript presents an in-depth analysis of the biophysical effects of land-use changes, based on simulations conducted with several Earth System Models (ESMs) participating in CMIP6-LUMIP. The results presented in the main text and supplementary material provide valuable insights into model responses to (mostly) increases in forest fraction. These increases result from either more intensive afforestation or reduced (avoided) deforestation under scenario SSP1 compared to SSP3, with the largest changes occurring in tropical Africa. In addition to the modeled changes in water fluxes and other key variables, the study computes several metrics designed to help understand the mechanisms driving changes in the surface water balance (P – ET).

We thank the reviewer for their comments and helpful suggestions. We have uploaded a revised version as well as a version with track changes. Below, we address each of the comments (original comments in black and answers in blue). We identify 4 main points from the "main comments" section of the review and respond to them individually at the end.

**Main comments:**

As noted, the study is comprehensive and, based on the model intercomparison, provides clear conclusions regarding robust changes in precipitation (P), evapotranspiration (ET), and consequently P – ET (Conclusion 1), as well as on the independence of land use-induced effects from the background climate (Conclusion 2; note that it seems unusual not to include a figure in the main text to support this conclusion).

However, regarding the mechanisms of change (Conclusion 3) — where this study invests more effort and could be more innovative — the authors, in my view, overcomplicate the analysis, overlook well-known causal chains, and fail to provide a credible explanation. Understanding the biophysical effects of land cover changes is clearly not straightforward. The change in a given variable depends on (1) the direct impact of surface forcing (i.e., changes in land surface properties), which can involve various processes (e.g., changes in radiative or turbulent fluxes that alter the surface energy balance), and (2) the atmospheric responses to (1). The atmospheric response is key, as it feeds back onto surface variables, either amplifying or damping the initial effect (e.g., changes in water recycling), and can export the impact beyond the region initially perturbed. The resulting net effect of, for example, afforestation, depends largely on the region, climate, and the spatial scale of the modified area, among other factors.

Mechanisms of change may be analyzed from the different optics (more typically modification in water or energy budgets either at the surface or the atmosphere). This paper focuses on the atmospheric water balance, with some useful simplification and decomposition of it terms.

Starting the results description, it reads (lines 211-212): "This study seeks to identify the mechanisms driving the Δ(P −E) pattern over the tropics from a dynamics perspective and to reconcile the apparent mismatch between the tropical Δ(P − E) and ΔNEI". It is not clear what the "apparent mismatch" refers to. Figure 2 shows a clear (and expected) response to tropical afforestation: increased ET and concomitant surface cooling. In turn, this change supplies moisture and latent heat to the atmosphere (Figures S4 and S10). Figure S4 also shows that the

increase in NEI is primarily due to latent heating, partially offset by other radiative effects. Why should we expect a different result in this case?

In several parts of Section 3, it is stated that dynamic effects in the lower troposphere dominate or explain the changes in P − ET (e.g., lines 238–239, 248, 293–294, 318–319), leading to Conclusion 3. As noted earlier, the atmospheric response is indeed key, but it does not explain the primary response of the models to tropical afforestation—namely, the increase in ET (leading to the reduction in P − ET). As shown by numerous previous studies—many of which are cited in the introduction—this increase in ET is a direct consequence of changes in surface properties such as increased LAI, canopy conductance, and turbulence. This pattern clearly dominates in this set of simulations.

This response is clearer during the dry season, as observed in central-southern Africa during the austral winter, where the change in P − E corresponds almost entirely to $\Delta$ET (Fig. 3). Naturally, a change rooted at the surface is then transmitted to the atmosphere, which can be analyzed through the water budget. In this case, increased ET leads to more humid air (Fig. S10), changes in atmospheric motion and moisture convergence, as illustrated by the omega approximation (Fig. 3f). However, this does not imply that changes in vertical motion and regional circulation are the primary causes of changes in P − ET, as the authors suggest; rather, these are atmospheric responses to surface forcing. I agree that the mechanisms discussed in the paper are relevant—particularly for explaining changes in P, when present—which may, in turn, modulate $\Delta$ET, but the explanation and conclusions should carefully follow a consistent causal chain and avoid reversing it.

Another interpretation that seem at least partially incorrect, yet presented as "true" throughout the paper, including in the conclusions and both abstracts, is that the reduction in (near-)surface wind is due to the drag effect of increased surface roughness. In contrast to the previous case, here the authors attribute an atmospheric response to afforestation entirely to a change in a surface property (i.e., roughness), without providing convincing evidence. While this should be a contributing factor, other well-known mechanisms could also contribute to—or even primarily drive—this response. One common mechanism involves temperature-induced changes in regional (monsoonal) circulation, which is completely overlooked in this case, despite all relevant indicators being present: a significant surface cooling and a concomitant sea-level pressure increase in central Africa (Fig. S10). Given that the mean pressure gradient and wind are directed toward the interior of the continent (Fig. 5), the resulting pressure increase would be expected to weaken the monsoonal circulation. The change in wind stress is not definitive evidence of the proposed mechanism, as it may instead result from changes in the low-level circulation. Moreover, the authors do not specify how wind stress was calculated.

These main comments affect a core conclusion of the paper, so the recommendation is for major revisions. However, all of the issues relate to the interpretation of results, many of which could be addressed through a re-assessment of the existing analyses.

We summarise the reviewer's main comments as follows and address them individually below:

1. The reviewer suggests including a figure in the main text to support Conclusion 2 regarding the independence of land use-induced effects from the background climate (paragraph 1).
2. The reviewer seeks clarification on the "apparent mismatch" between NEI and Δ(P − E) mentioned in the introduction and questions the expectation of a different result given the observed responses to tropical afforestation (paragraph 4).
3. The reviewer argues that the study primarily focuses on atmospheric dynamics as the primary driver of changes in P − ET (Conclusion 3). They assert that the increase in ET, a direct consequence of changes in surface properties, is the primary response to tropical afforestation and subsequently drives the reduction in P − ET (paragraphs 5-6).
4. The reviewer proposes an alternative or additional mechanism for the reduction in near-surface winds, suggesting that temperature-induced changes in regional (monsoonal) circulation, linked to surface cooling and pressure increases, should be considered alongside the drag effect of increased surface roughness (paragraph 7).
5. The reviewer requests clarification on the calculation of wind stress.

**Regarding main comment 1**: We understand the reviewer's comment but decided not to include Figure S3 in the main text since it shows very similar changes to Figure 2, which is included in the main text. In other words, there is minimal additional quantitative information in Figure S3 that isn't already included in Figure 2. We believe this is a good use case for a supplemental figure.

**Regarding main comment 2**: We appreciate the reviewer pointing out the lack of clarity regarding the "apparent mismatch" between the simulated Δ(P − E) and ΔNEI. Our statement refers to the expectation, based on established atmospheric dynamics, that an increase in Net Energy Input (NEI) over the tropics would typically drive a strengthening of the Hadley circulation and an overall increase in net precipitation (P−E) within the inner tropics. We include below a schematic representation of the moisture and energy transport in the Hadley circulation (Figure R1).

[Figure]

*Fig. R1: Schematic illustration of the energy transport (red arrows) and moisture transport (blue arrows) in the overturning circulation in the tropics.*

However, our simulations of tropical afforestation show an *increase* in NEI (primarily due to enhanced latent heat flux from the afforested areas) alongside a *reduction* in P−E in large areas of the inner tropics. This is the "apparent mismatch" we aimed to reconcile. While the southward shift and contraction of the ITCZ we observed are consistent with energetic frameworks linking NEI and ITCZ position (e.g. Byrne and Schneider 2016), as mentioned on Lines 208-210, the overall drying (P−E reduction) is not the dynamically expected response to a positive NEI anomaly in the tropics.

The reviewer correctly points out the increased ET and surface cooling, which supply moisture and latent heat to the atmosphere, contributing to the positive NEI. Our analysis delves into why this increased energy input does *not* translate into increased net precipitation in the core tropical region in our simulations, focusing on the role of altered moisture convergence patterns despite the enhanced energy.

To better clarify this, we have edited the text in the result section as follows (L. 203-210): "However, this response seems counterintuitive from an atmospheric dynamics perspective: Despite a simulated increase in NEI (given by the sum of the surface latent heat, surface sensible heat and net radiative energy into the atmospheric column; Eq. 6) over the tropics due to afforestation (Fig. 2e, S4), we observe an overall reduction in net precipitation. Based on atmospheric dynamics, we would expect that an increase in tropical NEI implies a strengthening of the overturning circulation and an increase in net precipitation throughout the inner tropics. While the southward shift and contraction towards the equator of the ITCZ are consistent with energetic frameworks based on NEI (Byrne and Schneider, 2014; Byrne and Schneider, 2016), the overall reduction of P−E is not."

Additionally, we changed L. 218-219 to "This study thus seeks to identify the mechanisms driving the Δ(P − E) pattern over the tropics from a dynamics perspective and to reconcile why the increase in NEI does not lead to the dynamically expected increase in (P − E)."

**Regarding main comment 3**: We fully agree with the reviewer that changes in the land surface properties following afforestation lead to direct changes in evapotranspiration.

Our focus in analyzing the dynamic effects (leading to Conclusion 3) stems from the question of *how* the increased evapotranspiration translates into the observed spatial patterns of Δ(P − E). We would like to clarify that if evapotranspiration would increase without any accompanying changes in atmospheric circulation, then precipitation would increase by the same magnitude through local recycling, resulting in net-zero changes in (P−E). This is true unless there is also a change in non-locally sourced precipitation, as would be needed to maintain a constant recycling ratio. A change in non-locally sourced precipitation requires a change in moisture-flux convergence, changes in which have been shown to be overwhelmingly driven by changes in atmospheric circulation (Chadwick et al. 2013; Wills et al. 2016; Fig. S9). Thus, to explain any non-zero changes in Δ(P − E), there must be changes in atmospheric circulation that redistribute moisture and alter precipitation patterns. Thus, while the initial change in evapotranspiration is land-driven, the resulting pattern of Δ(P − E) has to be caused by changes in atmospheric circulation. This necessitates a dynamics-focused analysis which we perform in Section 3.

We have added the following sentences to the introduction for clarification (L. 69-74): "Critically, if evapotranspiration were to increase without any accompanying changes in atmospheric circulation, precipitation would increase by the same magnitude through local recycling, resulting in net-zero changes in P − E. Therefore, to explain any non-zero changes in P − E, there must be changes in atmospheric circulation that redistribute moisture and alter precipitation patterns. Thus, changes in P − E are closely linked to atmospheric circulation changes, which drive large-scale redistribution of moisture and energy (e.g. Seager et al., 2010; Ma and Xie, 2013; Chadwick et al., 2013; Wills et al., 2016)."

Additionally, we also added the following explanation to the result section (L. 237-238): "Note that the non-zero changes in P−E imply that the afforestation in this region leads to changes in atmospheric circulation and moisture transport."

**Regarding main comment 4**: We thank the reviewer for this comment. Indeed, the thermodynamic/energetic influences on the monsoon are an important factor in addition to the surface roughness changes, and we have already investigated it extensively in the manuscript, albeit not quite in the way the reviewer is describing. It is not near-surface temperature over land that determines the strength of the monsoon, but near-surface moist static energy (or moist entropy), as has been extensively documented in the literature (Emanuel 1995; Geen et al. 2020; Harrop, Lu and Leung 2019; Ma et al 2019). We have now added two additional subpanels to Figure S10 showing the increase in near-surface moist static energy over central Africa (reproduced below), which show that the near-surface MSE actually increases. This is opposite of the change in temperature due to the large increase in near-surface specific humidity. Therefore, this mechanism would actually weaken the monsoon. This perspective based on near-surface MSE is a complementary perspective to the perspective based on NEI discussed throughout the main text and summarized in the schematic (Figure 8). Relating back to the reviewer's main comment 2, this is another way of explaining why the P-E change is counterintuitive, because it is opposite to the change in the monsoon that would be inferred from the change in near-surface MSE alone. We have added a sentence explaining that the same conclusions are reached when using near-surface MSE instead of NEI to investigate the energetic influences on the monsoon (L. 283-284).

[Figure]

*Fig S10: Spatial maps of changes in (a, e) surface specific humidity $\Delta q_s$, (b, f) near-surface temperature $\Delta Tas$, (c, g) sea level pressure $\Delta$ SLP and (d, h) near-surface moist static energy $\Delta MSE_{sfc}$ in MJJAS (upper row) and NDJFM (lower row) over Africa in AFFOREST. Contours show the BASE values with a spacing of 2.5 g kg$^{-1}$ for $q_s$, 3°C for Tas, and 2.5 hPa for SLP. Pink contours enclose the regions where there are at least 5\% of afforestation. Stippling shows where 6 out of 7 ESMs agree on the sign of change.*

As for the change in surface pressure, this is influenced by both mechanisms, because the link between anticyclonic motion and divergence operates through the near-surface vorticity balance, where wind-stress curl is proportional to mass convergence. The implications of this balance for tropical circulations and P-E is discussed extensively in Section 4 of Wills and Schneider 2015. Trees have a large influence on the surface roughness, which determines the relationship between the near-surface wind and surface wind stress, so the quantitative relationship between anticyclonic motion (as evident in SLP) and divergence will be modified by this change in surface roughness.

References:
- Emanuel, K. A. (1995). On thermally direct circulations in moist atmospheres. *Journal of the atmospheric sciences*, *52*(9), 1529-1534.
- Geen, R., Bordoni, S., Battisti, D. S., & Hui, K. (2020). Monsoons, ITCZs, and the concept of the global monsoon. *Reviews of Geophysics*, *58*(4), e2020RG000700.
- Harrop, B. E., Lu, J., & Leung, L. R. (2019). Sub-cloud moist entropy curvature as a predictor for changes in the seasonal cycle of tropical precipitation. *Climate Dynamics*, *53*, 3463-3479.
- Ma, D., Sobel, A. H., Kuang, Z., Singh, M. S., & Nie, J. (2019). A moist entropy budget view of the South Asian summer monsoon onset. *Geophysical Research Letters*, *46*(8), 4476-4484.

**Regarding main comment 5:** The wind stress is calculated as the magnitude of the eastward and northward wind stress (tauu and tauv) both of which are standard outputs from CMIP6.

**Some specific comments:**

- Lines 37–39: Canopy conductance/resistance is also a key factor.

  Thanks, we changed the sentence to "However, trees also enhance evapotranspiration through their larger leaf area and deeper root systems (Bonan, 2008), physiological control of transpiration through canopy conductance, as well as through the enhancement of turbulent fluxes by their influence on surface roughness." (L. 37-39)

- Line 66: Runoff is defined locally (or in a grid cell in a model). The integrated runoff over a basin leads to river streamflow.

  We have removed the parentheses after runoff in L. 67 to prevent confusion about the definition of runoff.

- Line 100: 1000 what? (Units are missing)

  Thanks for spotting this, we have changed it to 1000 Mha.

- Definitions in Section 2.2.1 (and throughout the text): Moisture, wind, and vertical velocity are, by definition, zero at the surface. These quantities must therefore be near-surface values. What level do they correspond to — 2 m, 10 m, or the lowest atmospheric model level? This is particularly relevant for the wind-based metrics used in the paper.

  We thank the reviewer for the comment. We have now specified for $q_s$ and $\boldsymbol{u}_s$ that we mean near-surface values (e.g., changes in L. 168-170, 263, 268, 303, 308, Caption of figure 5-7). We have also added the following sentence "Note that near-surface temperature and humidity variables are defined at a height of 2 m, while near-surface wind variables are defined at 10 m, consistent with cmorized CMIP6 variables." (L. 168-170).

  Please also note that the near-surface vertical wind does not only come from convergence below 10 meters, but also due to flow parallel to the surface when the surface is slanted in pressure coordinates.

- Explicitly state pressure vertical velocity throughout the text, as it is omitted in several sections.

  Changed.

- Line 171: The phrase "three-dimensional" is unnecessary here.

  Changed.

- Lines 195–196: P – E defines surface runoff.

  We wrote that net precipitation "...is a key control of surface runoff" (L. 199-200) since there are other factors like the exchange with ground water or soil water storage which impact surface runoff.

- Line 258: Again, there is no such thing as "surface vertical wind". The level used for near-surface circulation analysis must be clearly defined.

  We change it to "near-surface vertical wind" and have now specified the levels at which the wind-related variables are defined in Section 2.2.1.

- Lines 263–266 and Fig. 5: This is very confusing. Why not directly use the pressure levels provided in the model outputs?

In order to illustrate the impact on surface values more clearly, we are interpolating the vertical velocity to sigma coordinates. In pressure coordinates, the fixed pressure levels represent different heights above the surface. In contrast, in sigma coordinates sigma = 1 is always at the surface which helps to understand the afforestation impact on near-surface winds. We opted to show the sigma coordinates in units of pressure to help the reader intuitively understand around which height in the atmosphere this relates to as well as to relate back to the pressure level determined to be relevant when looking at the omega scaling (around 700 hPa).

In order to explain this better we have edited the text as follows (L. 273-277): "Note that we interpolated the pressure vertical velocity profiles to sigma coordinates (defined as pressure normalized by the grid-cell surface pressure in the BASE dataset). This coordinate transformation was performed to provide a clearer representation of effects on surface values, as pressure coordinate surfaces vary in their height above the surface, while in sigma coordinates $\sigma = 1$ always corresponds to the surface itself. The resulting sigma coordinates are presented in units of pressure by multiplying by the area-averaged surface pressure from the BASE dataset."

- Lines 422–424: I agree that having a large model ensemble allows for more robust conclusions, but model differences are also of great interest. The authors could elaborate more on this in the discussion.

  We have added the following sentence to the discussion: "While our large ensemble strengthens the robustness of our overall conclusions, the inter-model differences in the simulated hydroclimate and circulation responses also offer valuable insights into the uncertainties and complexities inherent in climate projections (Fig. S7, S8)." (L. 436-438)

- As noted in the manuscript, a large ensemble of simulations also allows for an increased signal-to-noise ratio. Yet, although scientifically relevant, the signal may not be particularly significant from the perspective of its impact on natural or human systems. In this sense, the authors could further discuss the intensity of the projected changes and their implications for, e.g, water availability, temperature, etc.

  We have calculated the fractional changes (AFFOREST/BASE) for evapotranspiration and precipitation and now mention the percentage changes in the text (L. 357-361 and Fig. R2). We further discuss qualitatively the potential impact for water resource management (L. 453-455) and on flooding (L. 455-457). However, we decided to make no additional quantitative statements about these effects since it will be highly dependent on the scale of afforestation / deforestation in these regions in the future.

[Figure]

*Fig. R2: Zonal-mean fractional changes in ΔP and ΔE over Africa (average over land area 20°W to 50°E). Stippling shows where 6 out of 7 ESMs agree on the sign of change. Note that we did not include the Δ(P-E) subplot as it saturates in all regions where the climatological P-E is near zero.*

- Figure S6: This figure shows absolute values (not changes), correct? If so, the delta symbol should be omitted.

  Thanks for noting this, we changed the figure caption and colorbar labels.

---

## Author Comment (AC2)

**Comments on "Mechanistic insights into tropical circulation and hydroclimate responses to future forest cover change" by Fahrenbach et al.**

If the symbol Δ refers to different states in forest cover, this study seeks to identify the mechanisms driving the Δ(P −E) pattern over the tropics from a dynamics perspective and to reconcile the apparent mismatch between the tropical Δ(P −E) and ΔNEI, where P is precipitation, E is evaporation and NEI is net energy input into the atmosphere. The methodology is based on comparing future scenario simulations from seven multi-ensemble models participating in the Land Use Model Intercomparison Project (LUMIP).

The main finding are as follows, in which are most pronounced over Africa

1. Afforestation and avoided deforestation lead to a robust increase in precipitation and evapotranspiration, alongside widespread decreases in net precipitation (precipitation minus evaporation) in the tropics.

2. The tropical hydroclimate response to future forest cover change is largely independent of the background climate under low- and medium-warming scenarios.

3. The changes in net precipitation over Africa are driven by the competing effects of surface drag-induced reduction of lower-tropospheric winds and net energy input-induced strengthening of deep-convective upper-tropospheric circulations.

The text itself is exceptionally well written. The review part is particularly interesting. The emphasis is on mechanisms rather than on a simple description of differences between simulated climate.

We thank the reviewer for their comments and questions. Below, we address each of the comments (original comments in black and answers in blue):

**Main comments:**

1)      Why is the response stronger over Africa? Is there a signal over the Amazon?

We thank the reviewer for this question regarding the regional differences in the simulated response. The stronger response observed over Africa, particularly in precipitation (P), evaporation (E), and their difference (P-E) over land, is primarily attributed to the significantly larger land-use change forcing in this region compared to the Amazon. As shown in Figure 1, the forest cover difference between the scenarios is substantially greater over Africa. Specifically, while the afforestation in scenario S1 over time are quite similar for both the Amazon and Africa (Fig. S1a), scenario S3 predicts large-scale deforestation over Africa, whereas changes over the Amazon are significantly smaller (Fig. S1b). Consequently, the difference between the S1 and S3 scenarios reveals a small afforestation signal over the Amazon but a large-scale avoided deforestation signal over Africa (Figure S1c). This substantial land-use change forcing over Africa leads directly to a pronounced local signal in the hydrological cycle (Fig. 2-4).

Regarding the presence of a signal over the Amazon, the simulations indeed show a response, albeit smaller in magnitude compared to Africa (Fig. S5). Similar to the African response, a small increase in P and E is observed over the Amazonian regions where land-use change occurs. Concurrently, temperatures decrease over these regions. This indicates that while the differences in land-use change scenarios over the Amazon are less extensive than over Africa, it still shows a small but consistent local hydrological and temperature response. We have added the following sentence to the manuscript to mention this response (L.222-224): "Note that while the Amazon also shows a consistent hydrological and temperature response, it is smaller in magnitude, consistent with the less extensive forest cover difference between the S1 and S3 scenarios (Fig. 1, S1)."

2)      Why aren't the PBL processes considered? Where is the PBL top in Fig. 8?

We thank the reviewer for their question regarding the Planetary Boundary Layer (PBL) processes. In our study, convergence within the PBL is fundamental to the omega-scaling approach (Equation 3, Fig. 3f, 4f), and throughout the manuscript, we examine both convective and mechanical (turbulent) influences on convergence in the PBL (summary in Figure 8). While we agree regarding the importance of PBL processes, we were not able to *directly* analyse changes in the PBL height. This is primarily because the PBL height variable ("bldep") is not available for any model participating in the LUMIP simulations and since the models do not output data at a sufficiently high vertical resolution to calculate the PBL height.

However, an *approximate indication* of the PBL top can be inferred from Figure 5, namely as the height where the influence of surface roughness changes on wind anomalies diminishes. Specifically, the wind anomalies are no longer significantly influenced by the roughness change above 700-800 hPa, which can be considered an approximate representation of the PBL top in the context of these simulations.

---

## Author Response (AR2)

**Comments on "Mechanistic insights into tropical circulation and hydroclimate responses to future forest cover change" by Fahrenbach et al.**

After the first revision, the authors have implemented changes that improve the manuscript. However, they continue to present interpretations of model outputs and bold conclusions regarding the simulated response to afforestation that, in my opinion, are not fully supported by the interesting analyses provided.

We thank the reviewer for their comments and helpful suggestions. We have uploaded a revised version as well as a version with track changes. Below, we address each of the comments (original comments in black and answers in blue).

**First, in general, concerning the role of atmospheric circulation in the reduction of P - ET:**

Several statements throughout the manuscript indicate that the reduction in net precipitation is primarily driven by dynamical changes:

Abstract: "The increased surface roughness not only increases evaporation, but also surface momentum fluxes, thereby slowing near-surface winds and reducing the orographic net precipitation"

Short summary: "... reduce net precipitation (precipitation minus evaporation) in these regions, which determines water availability. This happens because trees slow near-surface winds, ..."

End of sect. 3.2.1: "Thus, the primary mechanism(s) explaining the net precipitation changes over Africa are of dynamic origin (i.e., related to changes in the time-mean circulation), rather than related to transient-eddy or thermodynamic changes."

Conclusion 3: "The changes in net precipitation over Africa are driven by the competing effects of surface drag-induced reduction of lower-tropospheric winds and net energy input-induced strengthening of deep-convective upper-tropospheric circulations."

While such mechanisms may be valid in explaining precipitation responses alone, they do not explain changes in P – ET, which is dominated by changes in evapotranspiration. The latter is primarily driven by altered surface properties due to afforestation, not by large-scale dynamic circulation changes. Thus, attributing changes in P – ET to dynamical processes misrepresents the underlying mechanisms and could mislead readers regarding the role of surface processes. In their responses and in a newly added paragraph in the revised manuscript, the authors state: "We would like to clarify that if evapotranspiration were to increase without any accompanying changes in atmospheric circulation, then precipitation would increase by the same magnitude through local recycling, resulting in net-zero changes in (P – E)."

Why should a change in ET automatically result in a precipitation increase of the same magnitude? This assertion is not supported by physical principles. While it is true that, to maintain water balance, increased ET should moisten the atmosphere, this does not necessarily lead to an increase in precipitation of the same magnitude. Even if the P/ET

recycling ratio remains constant, this does not imply that absolute changes in P and ET will be equal.

We thank the reviewer for asking for further clarification on the role of atmospheric circulation on the changes in P-E.

In a perfectly closed system with no horizontal moisture transport, an increase in E would lead to an equal increase in P over time. This happens because the water has nowhere else to go; while the air would first moisten, a new equilibrium would be reached where the higher precipitation rate balances the higher evaporation rate. The result would be a zero change in net precipitation (P-E). The only way to avoid a zero change in P-E is if atmospheric moisture transport, which is controlled by the atmospheric circulation, adjusts to modify the moisture transport; this is simply a statement of moisture conservation. To better explain this in the text, we have edited the above-mentioned sentence to (L69-73):

"Critically, if evapotranspiration were to increase without any accompanying changes in atmospheric **moisture transport**, precipitation would increase by the same magnitude through **conservation of moisture**, resulting in net-zero changes in (P - E). Therefore, to explain any non-zero changes in P - E, there must be changes in atmospheric moisture transport."

However, our simulations show a robust, non-zero change in P-E, which demonstrates that the system is not closed. This non-zero change is direct evidence that large-scale atmospheric circulation has responded to the initial increase in E by redistributing moisture. We agree with the reviewer that the initial change in E is a land-driven input. However, changes in P-E require considering also how the atmospheric circulation and moisture transport adjust to this E change, since P-E is determined by moisture flux convergence through conservation of moisture. Thus, the moisture budget mechanisms discussed in the manuscript explain changes in P-E, not precipitation.

Importantly, our moisture-budget analysis shows that the P-E response is shaped by the dynamic circulation response. In particular, we show that the dynamic component of moisture flux changes,  $\Delta$ MCDyn, is the dominant contributor to the total  $\Delta$ (P-E) pattern over Africa. This provides direct, quantitative evidence that the net precipitation changes are not a passive consequence of changes in surface evapotranspiration but are instead a direct result of a dynamical atmospheric response.

In summary, while the increase in evapotranspiration is the initial physical forcing, it is the resulting change in atmospheric circulation that physically controls the spatial pattern and magnitude of the non-zero  $\Delta(P-E)$  response. Our focus on the dynamics is therefore a necessary step to explain the robust and physically constrained changes observed in net precipitation (P-E) across the models, and we do not attempt to quantify the relative contributions to evapotranspiration changes in isolation, as discussed more below.

**Second, with regard to the drag effect on evapotranspiration:**

As shown by several studies, increased surface roughness is only one of several mechanisms through which forests can enhance turbulent fluxes. If this mechanism were dominant, we would expect both latent and sensible heat fluxes to increase due to reduced aerodynamic resistance. However, in these simulations, there is a clear increase in latent heat flux, while the response of sensible heat flux is weak and of different sign (Figure S4). This pattern—a decreasing Bowen ratio—suggests that higher ET is primarily driven by changes in surface plant properties such as canopy conductance, rooting depth, and LAI, rather than by increased roughness alone.

We agree with the reviewer that evapotranspiration changes can be related to several different changes in surface properties, including deeper root system, physiological control of transpiration (canopy conductance) and surface roughness changes. Our study does not claim to disentangle these different influences on evaporation, and the simulation setup would not allow us to do so. Rather, it focuses on P-E, which as discussed in our response above is determined by the atmospheric moisture budget. Our moisture budget analysis demonstrates that a slowdown of near-surface wind convergence is a primary factor reducing P-E, as can be seen in the  $\Delta S$  term (Fig. 3, 4).

We believe that a sentence in the abstract might have mistakenly given the impression that we are saying surface roughness changes are the dominant influence on evaporation changes, and we have modified it such that it implicitly included changes in plant properties (which are further discussed in lines 36-38):

"Not only do forests increase evaporation, but they also increase surface momentum fluxes, thereby slowing near-surface winds and reducing orographic net precipitation."

**Third, with regard to the changes in circulation:**

Surface drag certainly affects near-surface winds, but it is not the only mechanism at play—changes in pressure gradients also play a key role. In their response, the authors acknowledge and partially agree a previous comment regarding changes in monsoonal circulation. However, this is not discussed in the revised text, and they ultimately maintain the same conclusions regarding the causes of circulation changes.

I invite the authors to consider a hypothetical numerical experiment in which surface roughness is held constant, while still allowing for other afforestation-driven changes (e.g., increased ET and alterations in the surface energy balance). Would we not still expect changes in surface temperature—and consequently in pressure and wind fields—somehow similar to those shown in the current model simulations?

We thank the reviewer for addressing the role of pressure gradients. As we mentioned in our previous response, the link between surface pressure and circulation is itself modified by changes in surface drag. This is because near-surface anticyclonic motion and divergence

operate through the near-surface vorticity balance, where wind stress curl is proportional to mass convergence. The direct influence of trees on surface roughness alters this relationship, meaning that the quantitative link between surface pressure changes and moisture divergence is itself changed by forest-induced changes in surface roughness. Thus, our manuscript discusses the influences on the monsoon circulation based on the moist static energy budget. Critically, it already captures the mechanism discussed by the reviewer, where near-surface cooling slows the circulation; however, this mechanism is overwhelmed by near-surface moistening, as shown in Fig. S10.

The reviewer's thought experiment—where surface roughness is held constant while other surface energy balance changes can take place—would of course also induce changes in the circulation. However, it would only capture the energetic effect, while the "full" afforestation scenario in our simulations changes both the surface momentum balance and the surface energy balance. Our key finding is that these two changes in the surface property changes trigger two separate competing effects, namely the surface drag and energetic effect. Thus, while in the hypothetical experiment changes in surface temperature and wind fields would occur, they would not show the same changes as the ones analysed in our manuscript, as it is missing the second, important effect due to surface roughness changes. They would only see the strengthening of the circulation that would be induced by the increase in net energy input (Figs. 6b, 7b), or equivalently the increase in near-surface moist static energy (Fig. S10), whereas the slowing of the near-surface winds due to the roughness increase would be missing.

---

## Author Response (AR3)

**Editor comments on "Mechanistic insights into tropical circulation and hydroclimate responses to future forest cover change" by Fahrenbach et al.**

The paper presents a novel analysis on how tropical circulation changes can explain changes in the net precipitation over Africa under climate change scenarios. The paper argues for a different control between changes in surface drag induced reductions in moisture convergence, versus increases in evaporation due to increases in latent heat due to afforestation. Authors find that changes are largely independent of the background climate under low and medium warming scenarios.

I find your paper a very interesting and original piece of work that might help to understand that changes in the P-E due to afforestation might be mainly driven by surface drag as opposed to other processes associated to physiology like roots, LAI, etc that increase Evaporation but may increase precipitation in a similar amount. Same as with one the previous reviewer, I still remain unconvinced by your claims on being surface drag the main driver of changes in P-E over certain regions and areas (what you call Western and Southern Africa in MJJAS), but I believe that even simple back-of-the-envelope calculations could go a long way in convincing readers that this is indeed the case in your simulations. Please see my major comments below.

We thank the editor for taking the time to read our manuscript in detail, and for his comments and helpful suggestions to improve our manuscript. We have uploaded a revised version as well as a version with track changes. Below, we address each of the comments (original comments in black and answers in blue). Additionally, we did another minor round of editing in the text.

**Major comments**

Line 246: "Transient eddy changes are also smaller than dynamic DMC changes ..." I am not entirely convinced this is true for continental regions. For instance, in Figure 3.e transient eddy changes in western equatorial Africa are large and opposite as changes in DMC over the same region, consistent perhaps with small changes in D(P-E) in Fig. 3c over the same region. Similarly, changes in DTE in Southeastern tropical Africa are higher than changes in DMC over that region, and consistent with positive changes in D(P-E). Certainly over the ocean near that ITCZ DMC dominates clearly over DTE. To what extent the fact that the TE changes are comparable with the MC changes, modifies the analysis, in particular, how appropriate is then to consider that all changes can be approximated by the changes that come from Wills and Schneider scaling? Please be very clear as to how the further analysis in the paper relies on DMC being the dominant change.

We thank the editor for his comment. We agree that in western equatorial Africa dTE is locally comparable in magnitude to dMC (Fig. 3e). However, due to the opposite signs of dTE and dMC this leads to net zero changes in d(P-E). This counterbalancing effect is important, showing that while the absolute magnitude of dTE is comparable to dMC, this effect does not show up in the net precipitation pattern. Similarly, we agree on the dTE contribution to the drying in southern Africa, a feature we had already highlighted in the original manuscript text. Nonetheless, our analysis shows that the primary mechanism determining the distinct features of the overall P-E pattern – specifically the drying over western and southern Africa and wettening over central Africa – is related to the dMC term in MJJAS and NDJFM. To better point out the nuanced effects

of dTE and the main contribution of dMC to the net precipitation pattern, we have edited Lines 246-253 as follows:

"The transient-eddy changes are generally smaller in magnitude than the mean-circulation-related changes and do not primarily determine the overall pattern across Africa. However, changes can be locally important, such as the contribution to the net precipitation drying in southeastern Africa in MJJAS (Fig. 3e) and wettening over central Africa in NDJFM (Fig. 4e). Furthermore, in specific regions like western equatorial Africa, large and opposing  $\Delta TE$  and  $\Delta MC$  anomalies can result in near-zero net change (Fig. 3c-e), highlighting local compensations. Nonetheless, the primary mechanism(s) determining the structure of the overall pattern in  $\Delta (P-E)$  over Africa are of dynamic origin (i.e., related to changes in the time-mean circulation), rather than related to transient-eddy or thermodynamic changes.

Regarding the empirical omega scale derived from Wills and Schneider 2015 and 2016: This dynamic scaling remains appropriate for our analysis as the dMC term is the most important influence on the d(P-E) pattern as discussed above. This scaling then gives an approximation of the dMC term, separately of changes in dTE. In Lines 160-162, we mention that "this approximation gives only a qualitative description of  $\Delta(P-E)$  since it neglects the effect of transient eddies and horizontal advection, and is not defined in regions where the surface extends above 700 hPa". However, since this scaling captures the main large-scale features of the dMC and thus d(P-E) pattern with the dry-wet-dry dipole across Africa in both seasons justifies these simplifications and confirms the utility of the scaling for isolating the main physical mechanism.

Lines 333-334: "Surface drag effect is the dominant ..." The wording here implies that we have a large confidence that the surface drag is the main driver of the decrease in P-E which goes back to the main concern of the previous reviewer.

We thank the editor for his comment. We used the term "dominant driver" as our analysis demonstrates that the surface drag effect has a large influence on the P-E pattern. By analysing surface drag-induced changes in P-E through changes in the surface term dS (Fig. 3f, 4f), we find a large reduction in coastal regions of Africa and large model agreement regarding this effect. This strong signal is clearly linked to the afforestation-induced increase in surface drag, which reduces near-surface wind speed, reduces the near-surface vertical velocity (Fig 5) and subsequently leads to a reduction of topographically-induced P-E.

However, in order to better point out what we mean by the term "dominant driver" as well as to underline the modulation of the surface drag and energetic effects by other, concurrent changes, we have edited Lines 337-343 as follows: "Both the energetic and the surface-drag effect are active over the entire African domain to afforestation, however the resulting net precipitation response depends on which effect is dominant (i.e., has the larger influence on vertical velocity at 700 hPa and thus P-E) over a certain region. The surface drag effect is the dominant driver of the net precipitation decrease over western and southeastern Africa during MJJAS and NDJFM, although the magnitude of change is modulated by the energetic effect. Conversely, the energetic effect is the dominant driver of the net precipitation increase over central Africa, though the surface drag effect still influences near-surface pressure vertical velocity."

Again in lines 260-265 the main decrease of the DS term along the West and East coast of Africa is attributed to an increase of surface drag due to afforestation, however, and as pointed out previously by one reviewer, the decrease in the flow might be due to the increase in latent heating over the continental region which tends to produce a decrease in net precipitation. Surely part of this decrease might have to do with changes in the surface drag, however, part of this decrease might be entirely due to other changes in the plant physiology that lead to a higher latent heat in the simulations, a net precipitation decrease and therefore a weaker monsoon circulation towards the continent with the concomitant decrease in the across isobar flow near the surface. Lacking a dedicated simulation that controls for changes are exclusively due to surface drag, I fail to see how the authors can claim a preferred way in which this effect reduces surface convergence. Again, the fact that other people: Bell et al, Lague et al, have not been able to make clean simulation changes to isolate this effect, does not preclude the authors to be clear on how they think such simulation should be conducted in order to isolate the effect, and what would be the results if their preferred mechanism is to be found. This can help others to clarify this point, which does not seem so far clear neither in the present paper nor in the existing literature.

The diagnostics already presented in the supplementary material—such as the inland increase in sea-level pressure, the rise in near-surface specific humidity, and the gain in moist static energy—are entirely consistent with the interpretation that drag weakens low-level convergence and precipitation, even under a more favorable thermodynamic environment. This provides a solid basis for your argument, and highlighting it clearly will strengthen the paper. In contrast, the evidence suggests that physiology, albedo, and roots primarily raise evaporation while simultaneously enhancing precipitation through recycling, leaving only a minor residual effect. If I am understanding the paper correctly, making this distinction explicit, and clarifying that you are using "drag" strictly in the aerodynamic sense of momentum sink and reduced aerodynamic resistance, will help sharpen the main message. In short, I encourage you to reinforce this interesting and important conclusion with the quantitative and diagnostic evidence you already have at hand, as doing so will greatly increase the impact and persuasiveness of your manuscript.

We thank the editor for his encouragement to clarify the evidence supporting our conclusion that the decrease in P – E via the dS term is primarily driven by an increase in aerodynamic surface drag due to afforestation, while also highlighting the consistency of the pressure and moist static energy changes with our existing mechanistic explanations using NEI.

We concur that the presented diagnostics are highly consistent with the aerodynamic drag interpretation. The inland increase in sea-level pressure (SLP) (Figure S10c, g) implies a weakened near-surface circulation despite a more favorable thermodynamic environment for monsoonal circulations provided by the increase in MSE (Figure S10d, h), which results from the rise in near-surface specific humidity (Fig. S10a, e) despite a broad scale cooling (Fig. S10b, f). This implies that there must be an additional influence on the near-surface circulation, because the thermodynamic environment would lead to an increase in circulation strength and P – E, which is not what is found. This can be understood based on our analysis in Figs. 5, 6c-e, and 7c-e showing that mechanical, dynamic weakening of low-level winds and convergence by increased aerodynamic drag is overpowering the opposing, positive influence of the thermodynamic environment (i.e., higher near-surface MSE/NEI/latent heat). The reduction in flow occurs despite the thermodynamically favorable latent heat increase, which is supporting our conclusions. We have incorporated this more explicitly by adding the following text (Line 283-

287): "The overall decrease in net precipitation occurs despite a thermodynamically more favorable environment for precipitation, marked by an increase in near-surface specific humidity and MSE (Fig. S10). The influence of the thermodynamic environment will be investigated with an atmospheric energy budget perspective in the next section, but the fact that the net precipitation decrease still occurs suggests the importance of the mechanical weakening in opposing the influence of the latent heating and near-surface MSE changes."

We have also clarified our methodology to explicitly state that we are referring to aerodynamic drag, which is strictly the momentum sink term resulting from changes in surface roughness and wind speed. We have added the following sentence to the method section (Line 167-169): "In our results, the change in  $\Delta S$  primarily reflects changes in aerodynamic drag on the near-surface wind ( $u_s$ ) caused by the alteration of surface roughness."

We agree with the editor that a dedicated, idealized simulation would be the definitive way to isolate the pure surface aerodynamic drag effect. Previous studies which tried to examine the impact of surface roughness alone (as in studies like Bell et al. (2015) and Lague et al. (2019)) changed the vegetation height, which simultaneously modifies momentum, heat, and moisture fluxes. Even in simulations where only the roughness length parameter in the surface bulk formula is changed (rather than the vegetation height as a proxy) this would lead to changes in evaporation. This can be explained by changes in the roughness length for momentum  $(z_{0m})$ typically being coupled with changes in the roughness lengths for heat and moisture ( $z_{0h}$  and  $z_{0a}$ ), thereby altering latent heat flux. Thus, we propose that the ideal conceptual simulation to isolate the pure aerodynamic drag (momentum sink) effect would be one where  $z_{0m}$  is perturbed while keeping the roughness lengths for heat and moisture  $(z_{0h}$  and  $z_{0q})$  fixed at their original values. This methodology leverages the distinct parameterizations for momentum and scalar transfer. By altering only  $z_{0m}$ , the change in near-surface wind speed is a pure momentum sink effect (aerodynamic drag), while the direct change in the aerodynamic resistance for evaporation (governed by  $z_{0a}$ ) is eliminated. The resulting thermodynamic changes would thus only be due to indirect effects (e.g., advection, large-scale circulation changes), allowing for a clearer focus on the primary role of the momentum-drag anomaly. We have now added this description of the proposed experiments to the discussion section (Lines 468-471): "Instead, we propose running simulations where the roughness length for momentum is modified independently of the roughness lengths for heat and moisture. This simulation setup would allow the isolation of the pure surface (aerodynamic) drag effect on near-surface wind speed from the aerodynamic influences on evaporation and sensible heat flux." We acknowledge that these idealised experiments are beyond the scope of our current analysis.

**Minor comments:**

Figure 6: Please check with some trusted source the name of the regions in Africa. West Equatorial Africa seems more appropriate than Central Africa, also East Central Africa, more appropriate than Southern Africa. Notice a typo in panels c,d and e in Vertical. Same in Figure 7. This can be a problem for the interpretation of the results as many will trust on the name you provide the regions on the abstract of the paper, without necessarily looking at the maps. Please make sure you are using the most appropriate name for each of the regions.

We have corrected the typo in Figure 6 and 7. We understand the editor's concern regarding the region names. However, since these regions in Africa are defined based on P-E anomalies rather than commonly used regions like for instance from the IPCC, there is no exact naming convention. Nonetheless, we tried to comply with the standard geographical naming of regions in Africa (e.g. Regions Of Africa), where central Africa is more located to the west of the continent. However, we have now renamed Southern Africa to Southeastern Africa to better describe this region (also with regards to the IPCC land regions ESAF and SEAF).

Line 170: cmorized, could be written as CMORized for better understanding?

Thanks, we changed it.

line 191: perhaps be more clear that all three processes mentioned lead to an increase in the latent heat flux.

We have reformulated the sentence (Line 189-191) to "In response to the afforestation, the zonal-mean evapotranspiration increases in the tropics due to enhanced transpiration through a larger leaf area, increased evaporative capacity due to deeper root systems and increased surface roughness (Fig. 2b; Bonan 2008)."

Line 341: "which undergoes" instead of "which is undergoes"

Thanks for spotting this mistake.